# Transcranial direct current stimulation and working memory: Comparison of effect on learning shapes and English letters

**Sriharsha Ramaraju**[1], **Mohammed A. Roula**[2]*, **Peter W. McCarthy**[3]

**1** Interdisciplinary Computing and Complex Bio Systems (ICOS) Faculty of Computing, Newcastle University, Newcastle Upon Tyne, United Kingdom, **2** Medical Electronics and Signal Processing Research Unit, University of South Wales, Treforest, United Kingdom, **3** Clinical Technology and Diagnostics Research Unit, University of South Wales, Treforest, United Kingdom

* ali.roula@southwales.ac.uk

**Data Availability Statement:** All relevant data are within the manuscript and its Supporting Information files.

## Abstract

We present the results of a study investigating whether there is an effect of Anodal-Transcranial Direct Current Stimulation (A-tDCS) on working memory (WM) performance. The relative effectiveness of A-tDCS on WM is investigated using a *2-back* test protocol using two commonly used memory visual stimuli (shapes and letters). In a double-blinded, randomised, crossover, sham-controlled experiment, real A-tDCS and sham A-tDCS were applied separately to the left dorsolateral prefrontal cortex (L-DLPFC) of twenty healthy subjects. There was a minimal interval of one week between sham and real A-tDCS sessions. For the letters based stimulus experiment, 2-back test recall accuracy was measured for a set of English letters (A-L) which were presented individually in a randomised order where each was separated by a blank interval. A similar *2-back* protocol was used for the shapes based stimuli experiment where instead of letters, a set of 12 geometric shapes were used. The working memory accuracy scores measured appeared to be significantly affected by memory stimulus type used and by the application of A-tDCS (repeated measures ANOVA $p<0.05$). A large effect size ($d = 0.98$) and statistical significance between sham and real A-tDCS WM scores ($p = 0.01$) was found when shapes were used as a visual testing stimulus, while low ($d = 0.38$) effect size and insignificant difference ($p = 0.15$) was found when letters were used. This results are important as they show that recollection different stimuli used in working memory can be affected differently by A-tDCS application. This highlights the importance of considering using multiple methods of WM testing when assessing the effectiveness of A-tDCS.

## Introduction

Working memory (WM) refers to the temporary storage and manipulation of information necessary for complex tasks such as language, executive function and long-term memory [1, 2].

**Funding:** SR University of South Wales centenary PhD Scholarship University of South Wales https://www.southwales.ac.uk/ No, The funders had no role in study design, data collection and analysis, decision to publish, or preparation of the manuscript.

**Competing interests:** The authors have declared that no competing interests exist.

WM dysfunction is observed in many neurological and psychiatric conditions such as stroke, trauma, schizophrenia, Alzheimer's, Parkinson's as well as in major depression [3–5] and ageing [6, 7]. Mnemonic encoding and extensive practice exercises may moderately improve WM in schizophrenia [8, 9], likewise antipsychotics might also improve cognitive functioning in schizophrenia [10, 11]. Sadly, the results of the above methods so far have been inconsistent [12]. As inconsistency in results might result from small variances in testing methodology and technique, it might also be important to look to the testing methods for WM as a potential source of previous inconsistency. It is crucial to further investigate this area, as any intervention showing capacity to improve WM would be of great interest to each of the geriatric, neurologic and psychiatric communities.

Neuroimaging studies demonstrated [13, 14] dorsolateral prefrontal cortex (DLPFC: Brodmann areas 9 and 46) involvement during WM tasks. Disrupting DLPFC activity using transcranial magnetic stimulation (TMS) leads to deterioration of WM performance [14–16], supporting a role for the DLPFC in WM.

Transcranial Direct Current Stimulation (tDCS) is a non-invasive brain stimulation technique considered capable of modulating spontaneous cortical activity. It uses low-intensity direct currents induced through a pair of rubber electrodes (covered by sponges) placed on the skin of the scalp [17]. The delivered currents are considered to induce polarity dependent changes in the cerebral cortex. A natural inducer of polarity dependent change, long-term potentiation (LTP) is an acknowledged model of the neural plasticity hypothesised to underlie learning and memory. Floel and Cohen [18] suggested that non-invasive cortical stimulation, in combination with memory training, might induce LTP.

Although, tDCS appears to have induced significant change in a few cognitive studies, inconsistencies still exist [19–21]. A growing body of literature [19, 21–23] suggests the importance that individual differences may have in moderating any influence that tDCS may have on cognitive functions [24].

Generally, anodal tDCS (A-tDCS) has been shown to have a positive effect on WM whereas cathodal tDCS (C-tDCS) has been shown to have no or negative effects. A number of human tDCS studies using anodal stimulation have reported enhancement in motor activity [25, 26], visio-motor activity [27], language learning [28], picture naming [29] as well as enhancement of WM [30, 31]. A 10-minute period of A-tDCS (1mA) applied to the left DLPFC (L-DLPFC) has been reported to enhance the performance of verbal WM tasks, when compared to a sham stimulation [2].

Regarding the range of current used, this generally varies from 1mA to 2mA [2, 19, 31, 32]. It would appear, at least in relation to WM in older adults [21], that there may be no statisitical difference between the effects of 1mA and 2mA. It has been suggested that any enhancement of excitability is dependent on stimulation duration as opposed to the strength (stimulation durtaion in turn drives the duration after effects) [33, 34]. However, longer durations might also cause redness on the scalp [35] and effect the double blinding capability. Although this a generalisation, we recognise that there may be exceptions where either higher or lower currents, or C-tDCS can be shown to have a positive effect on WM. In this study the choice was to be consistent with the majority of findings reported and use 1.5mA A-tDCS for 15mins.

A commonly used measure of WM performance is the *n-back* test, which has been shown to activate the DLPFC and posterior parietal cortex [36–41]. Funahashi et al. [42] studied the importance of DLPFC in the processing of stimuli and what happens to this activity during the retention period. Each half of the prefrontal cortex appears to be functionally specific, with right hemisphere (R) being involved in particular spatial WM tasks while left hemisphere appears vital for non-spatial WM tasks such as verbal WM tasks [42]. TMS [43] and lesion [44] studies confirmed the importance of L-DLPFC. These studies have shown that focal

damage and temporary disruption of L-DLPFC, but not R-DLPFC, related to impairment in WM task performance.

The *n-back* test is an active as it updates the WM continuously [45], and it has been used in prior tDCS studies [2, 30, 46]. Various forms of stimuli can be used in the *n-back* test: the most common are letters, shapes or numbers. It is usual for tDCS studies of WM to employ only one the form of a challenge; for example, a protocol might only use letters. tDCS has been used on other modalities as part of WM testing using the *n-back* protocol, these have included verbal tongue twisters [47] and shapes [41]. Single WM test protocols using a letter based *n-back* protocol with tDCS have shown that tDCS stimulation can increase performance in both healthy and neurologically compromised (Parkinsons disease) individuals [1, 31, 48]. Furthermore, a one week "wash-out" period is considered appropriate between sessions, in cross-over controlled trials of A-tDCS, to ensure little residual effect [20, 30].

To the author's knowledge, only one previous study has used multiple WM stimuli in the same experiment [19]. This study used both visual (shapes) and verbal (English letters A-J) stimuli, reporting an improvement in WM accuracy across both modalities in their population of educated adults. Although Berryhill and Jones [19] used both visual and verbal stimuli in their research, their study was aimed at observing the effects of A-tDCS on individuals of different education levels, and not to differentiate the effect of the stimulus on either of the forms of WM.

As mentioned above, different studies have used different stimulus to gauge the effect of tDCS on WM. However, none of them considered the effect of stimuli. The objective of our study is to investigate the effect of A-tDCS application on participants' performance on *2-back* WM tasks using two variations of memory stimuli, one involving recall of letters and the other involving recall of geometric shapes. We hypothesise that the real A-tDCS application group will have improved WM performance when compared to the sham group. Given recalling shapes and letters use different cognitive pathways and that recalling shapes is a relatively novel task in comparison to letters (used commonly in reading tasks), it is likely that recalling performance of random shapes has more potential for improvement than recalling familiar letters. We therefore also hypothesise the WM score improvement with shapes after A-tDCS will be more significant than with letters.

## Material and methods

### Participant selection

Twenty male subjects (aged 30±8 years) met the inclusion criteria which included not having diagnosed mental or physical health issues. All the participants were right handed. The participants gave a signed informed consent to participate. Although the study had been advertised without gender bias, only male subjects volunteered.

Subjects were recruited from the University of South Wales' student population comparable to that of Berryhill and Jones study [19]. Those who showed interest were given an information sheet about the experiment and further screened for a history of either neurological ailments or, taking of medication targeting or expected to affect their central nervous system. Subjects were requested to abstain from sugared, caffeinated or alcoholic drinks before the stimulation sessions. Before signing their informed consent, subjects were shown the equipment to be used and told exactly what would be expected of them. The study was approved through the ethical review process of the Faculty of Computing Engineering and Science at the University of South Wales.

## Anodal-tDCS (A-tDCS) application protocol

A double-blinded, randomised, cross-over sham-controlled protocol was used for this study. Subjects underwent two experimental tDCS sessions: one with sham A-tDCS (referred to as sham) and the other using real A-tDCS (referred to as tDCS).

The anodal electrode was placed over L-DLPFC and the cathode was placed on the right supra-orbital area (SO) corresponding to F3-Fp2, as per the 10–20 international system for EEG electrode placement. This montage is consistent with that used in previous research studies to investigate the effect of tDCS on WM [1, 2, 30].

The tDCS device (DC stimulator plus; neuroConn GmbH) delivered a low-intensity current to the brain using rubber electrodes (covered in sponge pads) of size, 5x7cm soaked in (0.9% NaCl) solution. The DC stimulator plus device has a "study mode" which is designed explicitly for the double-blind studies. NeuroConn provides two conditions of codes (sham and tDCS) that can be set to select which option is delivered.

The order of sham and tDCS presentation to each participant was based on a random generation (Microsoft Excel) of which code to use, resulting in 11 subjects receiving sham and 9 subjects receiving tDCS in their first session). An independent investigator gave two conditions of codes (determining the type of stimulation) to the researcher conducting the experiment who was unaware of which code was active tDCS. The researcher then entered the codes into the tDCS unit for each experiment, which was then performed. The second session was conducted using a complementary code so that each participant underwent one sham and one tDCS session without either the participant or the researcher knowing which one was which. The two sessions (sham and tDCS) were separated by at least one week.

Both the sham and tDCS sessions consisted of 15 minutes of stimulation [20, 49]. The sham session consisted of current ramping up to 1.5mA over a 8s period, followed by a 5s fade out and 870s without any significant stimulation (just impedance control). The tDCS stimulation consisted of current ramping up to 1.5mA over an eight second period, followed by continuous stimulation at 1.5mA. During the experiment, the impedance was always maintained less than the threshold value (12KOhm for 1.5mA) as per the recommendation of the manufacturers of the tDCS device. No adverse effects or complaints were received from subjects. During the stimulation, subjects were reading books, using mobile phone or resting. Offline stimulation was used in this study.

## Working memory measurement protocol

WM tests (shapes or letters) were applied separately, with the choice of which test was presented first being determined by random number generation. A single sham or tDCS session included two WM test runs (one "letters" and one "shapes"). For the duration of the experiment, subjects sat in an armless office chair, facing a computer monitor placed approximately at 0.7m in front of them at eye level (180˚) with their right index finger on the right arrow of the keyboard. Before the start of the experiment, the subjects were briefed on how the 2-back test for WM would be conducted and were given the opportunity to rehearse both the letters and shapes paradigms.

In the letters WM test, subjects were shown English letters (A-L) one at a time (each appearing for 2s) presented in a randomised order. A blank screen was presented for 1.5s between letters. The subjects were instructed to press the right arrow key on the keyboard if they recalled that the current letter was identical to the one seen two steps back, or doing nothing if they were considered not to be identical. The subjects were instructed to press the button anytime between presentations of a cue to end of 1.5s blank screen. In the case of the shapes 2-back test (slant s, oval, rectangle, mirrored tick mark, equilateral triangle, right angled triangle,

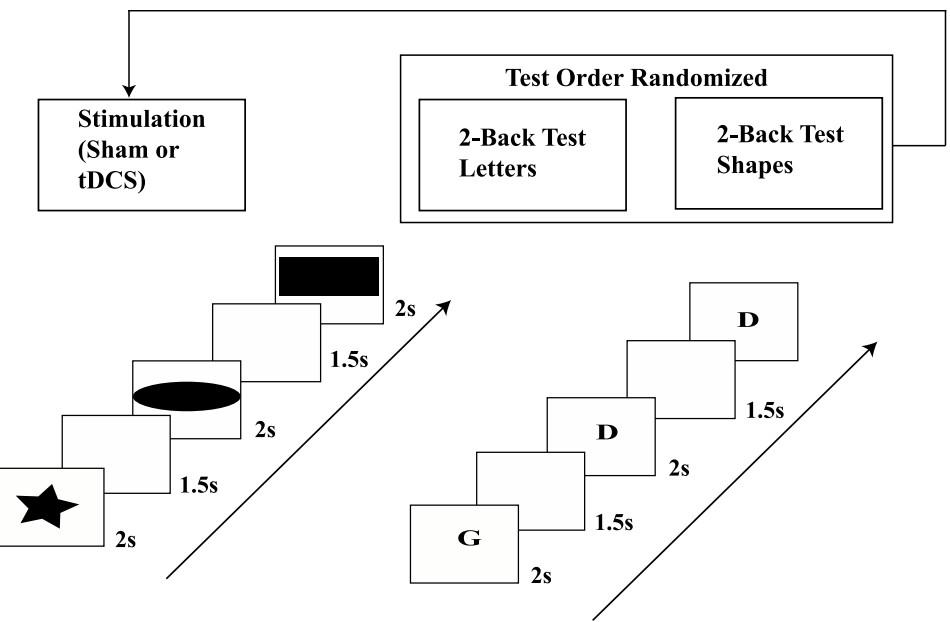

**Fig 1. The experimental protocol showing the sequence of the 2-back test using letters (L) and shapes (S).**

rhombus, pentagon, 4-sided star, 6-sided star, thunderbolt, inverted jig-saw), the procedure was the same as for the letters protocol, with shapes being presented instead of letters. Each subject was given a chance on the first day to rehearse (both alphabets and shapes test) only once before taking part in the actual test. The sequence of cues in practice test are different to the actual test.

In each session (sham or tDCS), subjects were presented with a total of two runs shapes and two runs letters separately, and each run consisting of 50 cues. Each cue displayed for 2s, and an inter-cue interval (inter-stimulus interval) of 1.5s. This adds up to $50*2+49*1.5 = 173.5$s (~3mins). After first run subjects were given a recover time of 15s followed by second run. Total time for a block: 3min+3min+15s = 6min 15s.

The numbers of 2-back matched pairs (targets) presented in session-1 and session-2 were 35 and 37 respectively. This slight disparity in number of targets was due to the selection being based on random number generation via Excel (Microsoft). The experimental procedure has been summarised in Fig 1. Prior to starting this experiment, the team ran a pilot study, to determine whether the subjects could perform the test by pressing a button when they recognised the appropriate symbol, in a similar manner to 0-back testing, without demand on memory. We did not formally test the sustained attention element, however as each test period for a single stimulus type was circa 3 minutes, attrition was not considered an important factor [50]. Furthermore, n-back with varied $n$ values (1, 2 and 3) was trialled to ascertain the optimal $n$ for the purposes of the experiment. For $n = 1$ accuracy was close to 100%, whereas $n = 3$ was difficult for most subjects with accuracy close to 0%. 2-back was found to provide the greatest range of accuracy results within and between subjects and was therefore selected for the main experiment.

## Statistical analysis

The working memory test answers were compared to actual correct answers to calculate true positives, true negatives, false positives, and, false negatives across stimulation and stimuli.

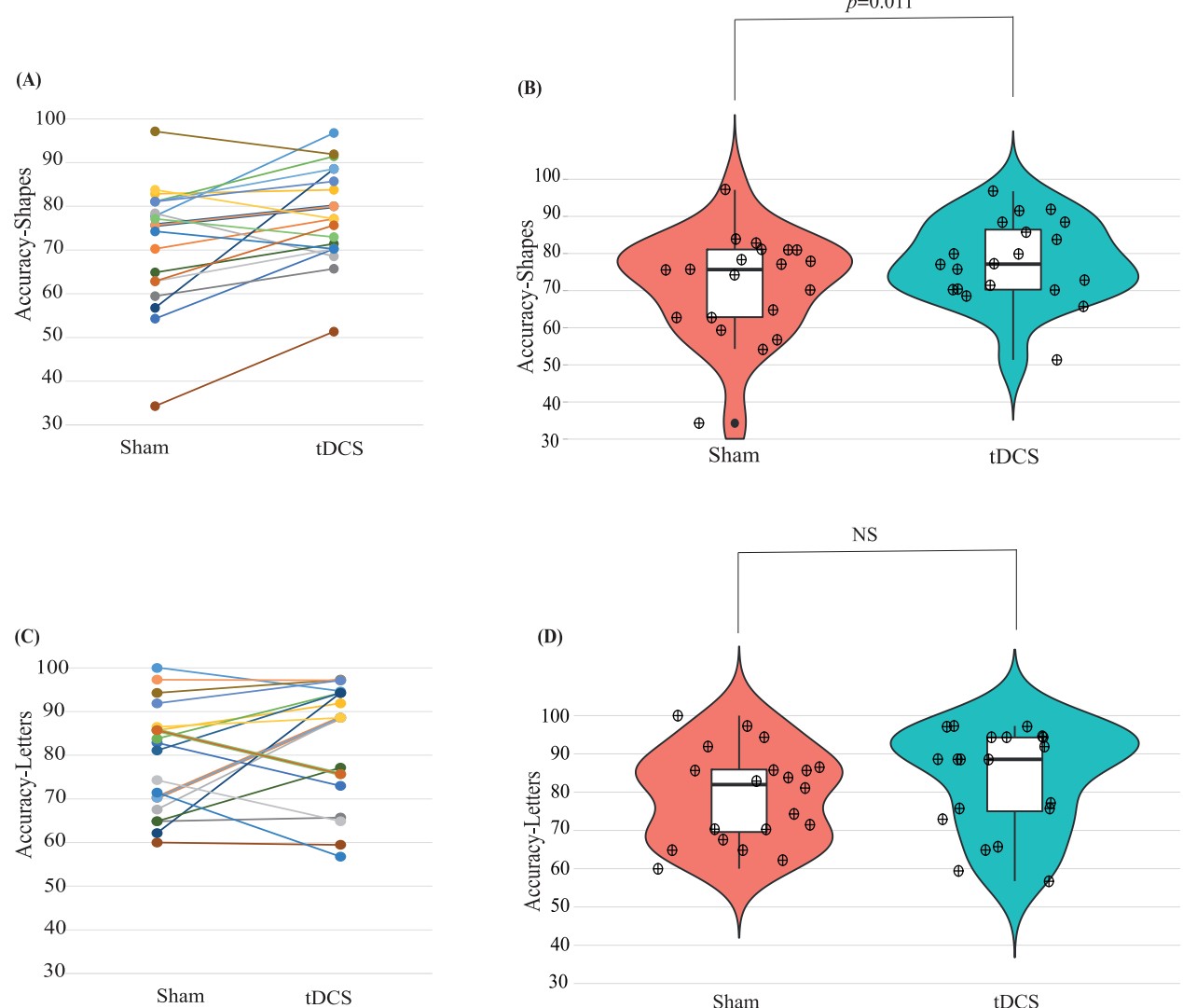

**Fig 2.** Transition plots of accuracies for 20 subjects across sham and tDCS conditions in (a) shapes and (c) letters. Violin plots (including box and scatter plots) across sham and tDCS conditions in (b) shapes (d) letters. NS-Non Significant ($p = 0.152$).

Accuracy which is calculated using (1) was used to determine the effect of stimulation and stimuli effects on WM and summarised in Fig 2.

$$\text{Accuracy} = \frac{\text{TP} + \text{TN}}{\text{TP} + \text{TN} + \text{FP} + \text{FN}} \qquad (1)$$

TP: True Positives, TN: True Negatives, FP: False Positives, FN: False Negatives.

Due to the nature of the study (cross-over sham-controlled trial design) a repeated measures ANOVA was used to quantify the combined effect of stimuli (stimuli: letters: 81.1±12.6, shapes: 74.7±12.5) and stimulation (stimulation: real: 80.5±12.2, sham: 74.2±12.5). To further understand the relationship, effect size and paired sample t-test were calculated. The relative degree of change (effect size) between sham and tDCS conditions across shapes and letters memory stimuli types was determined using Cohen's $d$-test.

## Results

### Effect of tDCS intervention on *2-back* test accuracy

Eighty percent of the subjects exhibited an increase in their WM accuracy on the shapes *n-back* test post A-tDCS (Fig 2(A) and 2(B)), compared to only 60% when using the letters-stimulus. This change is evident from both transition (Fig 2(C)) and violin plots (Fig 2(D)) plots. After the application of A-tDCS the left tail persists (Fig 2(D)) and this is opposite for the shapes accuracies (Fig 2(B)).

A significant change in *n-back* test accuracy between tDCS and Sham [$F(1,76) = 5.09$, $p = 0.02$, partial-$\eta^2 = 0.063$] as well as par-significant change across memory stimuli types [$F(1,76) = 3.41$, $p = 0.06$, partial-$\eta^2 = 0.043$] and insignificant interaction between stimuli and stimulation [$F(1,76) = 0.276$, $p = 0.7$, partial-$\eta^2 = 0.002$] was found. These results indicate the significant effect of tDCS stimulation on working memory accuracy irrespective of the type of stimuli used. This also indicates a par-significant difference in outcome between the stimuli, regardless of the type of stimulation used.

The above results were recomputed after removing the outlier subject from letters and shapes groups and the results seems to be unchanged.

Stimulation (sham and real): $F(1,76) = 3.91$, $p = 0.05$, partial $\eta^2 = 0.05$.

Stimuli (letters and alphabets): $F(1,76) = 5.17$, $p = 0.02$, partial $\eta^2 = 0.06$.

Interaction effect: $F(1,76) = 0.068$, $p = 0.8$, partial $\eta^2 = 0.001$.

Two-tail paired *t*-test between sham and tDCS across shapes: 0.02 (Cohen's $d = 0.56$).

Two-tail paired *t*-test between sham and tDCS across letters: 0.15 (Cohen's $d = 0.36$).

Cohen's *d*-test produced *d*-values (between sham and tDCS groups) of 0.98 and 0.38 for shapes and letters respectively indicating high and low effect sizes. A paired sample t-test between sham and tDCS in shapes stimulus resulted to be significant (t[19, 2.82], $p = 0.011$) whereas in letters stimulus resulted to insignificant (t[19, 1.48], $p = 0.15$).

## Discussion

The experiment outlined in this study investigated the effect of electrical stimulation on working memory, revealing large (WM scores for the recall of shapes: sham and tDCS) to medium (WM scores for the recall of letters: sham and tDCS) Cohen's *d*-values, indicating small (62%) to large (84%) overlapping [51] of the two distributions, respectively. This test also revealed that 84% of the A-tDCS condition is above the mean of sham condition when shapes stimulus is employed compared to 62% when letters are employed. In addition, it indicates that the effects of A-tDCS may be seen more easily when using the shapes version of the visually applied *n-back* test compared with the letters version and may help explain some of the inconsistency across the literature.

Our study is consistent with other work in this area [1, 2, 30], which reported significant improvements in WM performance post A-tDCS stimulation. This is consistent with promotion of long-term potentiation (LTP), where a short period of strong synaptic activation leads to a lasting increase in excitatory postsynaptic potentials. The difference in the effect sizes between letters and shapes might be due to variations in the spatial structures used for each of the stimulus.

A recent meta-analysis of motor and cognitive tDCS studies highlighted the difficulty in predicting the outcome of tDCS on behaviour [52]. One possible explanation might be due to the difficulty in detecting the change in WM performance when there is no deficit in the performance previously. The data presented in Fig 2 appears to support this statement. A further

issue with the letter task could be the lack of capacity for it show large changes. The average WM accuracy when the letter stimulus was used, or for that matter individual accuracies in sham stimulation, were at a high level already (Fig 2(C) and 2(D)). A possible reason for this observation could be that the WM task with letters stimuli may have been too easy for these participants.

Subjective confirmation of the above suggestion follows from asking subjects which of the two memory stimuli they felt harder to recall. Most of the subjects indicated it was the shapes rather than the letters. When asked why it was difficult to recall shapes, most of them answered that they could speak or shout letters in their minds loudly, but they could not do this with shapes. This observation has support from Smith et al. [39], who showed that visual WM can depend on the verbal encoding of visual stimuli. This could be limited capacity for the letters to show improvement, alternatively perceived ease might make the subject concentrate less and thus perform more poorly. A second possible explanation might be that cognitive processes such as encoding, maintenance, selection and decision making are the critical functions of DLPFC [1] and it might be that one of these functions is not working in tandem with the others while letters are being used as a stimulus. Alternatively, subjects might have found it difficult to encode the shapes when compared to letters in sham stimulation and application of A-tDCS helped circumvent the problem of critical functions and encoding. In this phenomenon, combining the repeated A-tDCS sessions with cognitive training appears to enhance WM [53, 54].

The aforementioned explanation may help to understand the learning rate for both shapes and letters in both the sessions. Daily usage (familiarity with) of English letters might have helped the subjects to learn quickly and significantly improve their accuracy in the second session regardless of A-tDCS. Whereas, the relative unfamiliarity with the shapes may have resulted in an increased difficulty to learn and remember them.

One potentially limiting aspect of the study was the absence of female subjects. The selection of a male only cohort, albeit due to chance, could also be considered a strength; reducing associated variables. It may be possible to extrapolate the general findings into the female population; however, introducing females into the study cohort might also have introduced a number of additional variables associated with the menstrual cycle, with its accompanying cyclical change in steroid hormones which are known to affect mood, concentration and other aspects of brain activity [55]. A another possible minor factor was that a small number of subjects returned for the second session at a different time of the day compared to their first session which may have affected their alertness during the WM tests and hence not allowed for controlling for variability within a day. Finally, it is important to point out that the study did not look at "reaction time" and its' interaction with accuracy and whether a ceiling effect [56] could have been affected by tDCS application.

## Conclusion

This paper has presented results showing the effect of A-tDCS on working memory to be dependent on memory stimuli used. Although a significant A-tDCS effect was found using the shapes-based WM stimuli, no such change was found for the letter-based test. This finding may have relevance in understanding the apparent selective effect of tDCS and its interaction with varied modes of brain activity. To better understand these findings, the functional connectivity of the working memory needs to be studied to determine the optimal use for A-tDCS. Response time was not considered in this study, but would be an interesting factor to consider in future comparative studies.

## Supporting information

**S1 Data. WM data.**
(XLSX)

## Author Contributions

**Conceptualization:** Sriharsha Ramaraju.

**Data curation:** Sriharsha Ramaraju.

**Formal analysis:** Sriharsha Ramaraju.

**Investigation:** Sriharsha Ramaraju.

**Software:** Sriharsha Ramaraju.

**Supervision:** Mohammed A. Roula, Peter W. McCarthy.

**Visualization:** Sriharsha Ramaraju.

**Writing – review & editing:** Mohammed A. Roula, Peter W. McCarthy.

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
