## [Decision Letter · Decision Letter 0]

9 Jan 2020

PONE-D-19-24574

­­Transcranial Direct Current Stimulation and Working Memory: Comparison of effect on Learning Shapes and English Letters

PLOS ONE

Dear Dr Ramaraju,

Thank you for submitting your manuscript to PLOS ONE. After careful consideration, we feel that it has merit but does not fully meet PLOS ONE’s publication criteria as it currently stands. Therefore, we invite you to submit a revised version of the manuscript that addresses the points raised during the review process.

We would appreciate receiving your revised manuscript by Feb 23 2020 11:59PM. To enhance the reproducibility of your results, we recommend that if applicable you deposit your laboratory protocols in protocols.io, where a protocol can be assigned its own identifier (DOI) such that it can be cited independently in the future. For instructions see: http://journals.plos.org/plosone/s/submission-guidelines#loc-laboratory-protocols

We look forward to receiving your revised manuscript.

Kind regards,

Tifei Yuan

Academic Editor

PLOS ONE

Journal Requirements:

Reviewers' comments:

Reviewer's Responses to Questions

**Comments to the Author**

1. Is the manuscript technically sound, and do the data support the conclusions?

Reviewer #1: Yes

2. Has the statistical analysis been performed appropriately and rigorously? 

Reviewer #1: Yes

3. Have the authors made all data underlying the findings in their manuscript fully available?

Reviewer #1: No

4. Is the manuscript presented in an intelligible fashion and written in standard English?

Reviewer #1: Yes

5. Review Comments to the Author

Reviewer #1: The authors employed twenty males to investigate the effect of A-tDCS on the left DLPFC when doing a working memory task (i.e., 2-back task, shape or letter). They stated that a significant A-tDCS effect was found in the shape working memory task not in the letter task. But the following issues should be addressed.

1. Participant Selection. More information should be provided (e.g., handness, and vision).

2. Working Memory Measurement Protocol. More details should be provided to improve its repeatability. When did the participants need to make a response? the onset of shape/letter? or the 2s-blank? Did you ask the participants to response as good and fast as possible? How about the practice phase before the formal test? One task had only one block? How long would the formal experiment take? How did you carry out “double-blinded”? Make these clear.

3. Results. More analyses should be added and corrected. The authors should add a separate part to describe the data analysis, not in the Results section. What’s the meaning of ‘TP’…in the Function 1? Please provide mean and SD/SE values in each condition. Add effect size (partial eta square) after p value when doing ANOVA. F(1,76,)should be F(1,76). Why not report the results of RT that is also important for n-back task? I expect to see the results of RT in the revision. I find one subject’s data is abnormal (below the chance level of 0.5) in the Fig 2A. Please provide the other version of the statistical results in the revision where you should delete this data point to find out if it affects the current results. The interaction effect was not significant, so why did you conduct a simple effect analysis by t-test? This doing seems not follow statistical requirements. The df and t values are wrong (t(38)=-2.81, p=0.011...).

4. Discussion. Not using 0-back as the control condition should be the other one limitation. As far as know, 0-back is the control condition for the classical n-back task. Why not use it in your study? Else, please discuss the influence of task difficulty between these two tests on your results.

5. Reference. The order of 54 and 55 is wrong.

6. Figure 1. The rectangle in the top left corner of Fig 1 seems to be unnecessary. The red line under the “tDCS” should be removed.

6. PLOS authors have the option to publish the peer review history of their article (what does this mean?). If published, this will include your full peer review and any attached files.

Reviewer #1: No

---

## [Author Response · Author response to Decision Letter 0]

23 Feb 2020

Response to Reviewer’s Comments

We thank the reviewer for the overall positive appraisal of the quality of the paper and the constructive feedback given. We are pleased to note that the reviewer found our paper technically sound, the data support the conclusion, the statistical analysis was appropriate and rigorous and the manuscript was intelligible and of good writing standard. 

Below, we show how we have addressed all the comments made by the reviewer and revised the paper accordingly. We have corrected all the errors and typos, added explanations as requested and clarified a couple points around methods. We believe the reviewers’ feedback has helped increased the quality and readability of the paper. 

Reviewer #1: 

The authors employed twenty males to investigate the effect of A-tDCS on the left DLPFC when doing a working memory task (i.e., 2-back task, shape or letter). They stated that a significant A-tDCS effect was found in the shape working memory task not in the letter task. But the following issues should be addressed.

1. Participant Selection. More information should be provided (e.g., handness, and vision).

Authors: We have taken on board this comment and have addressed it in Page 4, line 4. All the participants were right handed. 

2. Working Memory Measurement Protocol. More details should be provided to improve its repeatability. When did the participants need to make a response? the onset of shape/letter? or the 2s-blank? Did you ask the participants to response as good and fast as possible?

Authors: The aim of this study was not to measure the reaction time, so participants were told to respond anytime from the cue is displayed to the end of the 1.5s-blank period (before the starting of next cue). This is now explained in page 5, line 26-27. 

2.1 How about the practice phase before the formal test? 

Authors: Yes, we can confirm that every participant had a practice test (the sequence of random shapes/alphabets used in the practice are different to that of sequence used in experiment). This has been clarified now in page 5, line 31-33.

2.2 One task had only one block? How long would the formal experiment take? 

Authors: Each block contains two runs, each run consists of 50 cues, with each cue displayed for 2s, and an inter-cue interval (inter-stimulus interval) of 1.5s. This adds up to 50*2+49*1.5=173.5s (~3mins). After the first run, the subjects were given a recovery time of 15s followed by second run. Total time for a block: 3min+3min+15s= 6min 15s.

The above explanation has now been added to the Methods section (page 5, line 34-38).

2.3 How did you carry out “double-blinded”? Make these clear.

Authors: The below paragraph was already present in the original manuscript (Page 4, line 31-37)

“An independent investigator gave two conditions of codes (determining the type of stimulation) to the researcher conducting the experiment who was unaware of which code was active tDCS. The researcher then entered the codes into the tDCS unit for each experiment, which was then performed. The second session was conducted using a complementary code so that each participant underwent one sham and one tDCS session without either the participant or the researcher knowing which one was which. “

The above paragraph explains the process of double blinding where neither researcher nor participants knew which stimulation (real or sham) is being applied.

3. Results. More analyses should be added and corrected. The authors should add a separate part to describe the data analysis, not in the Results section. 

Authors: We have now added a “Statistical Analysis” section above the “Results” section in Manuscript.

3.1 What’s the meaning of ‘TP’…in the Function 1? 

Authors: This comment has now been addressed in Page 6, line 13.

3.2 Please provide mean and SD/SE values in each condition.

Authors: This comment has now been addressed in Page 6, line 16-17.

3.3 Add effect size (partial eta square) after p value when doing ANOVA. 

Authors: Partial eta square has now been added after p-value in two-way ANOVA results (page 6, line 33-36) as advised by the reviewer.

3.4 F(1,76,)should be F(1,76). 

Authors: This comment has now been addressed in Page 6, line 33.

3.5 Why not report the results of RT that is also important for n-back task? I expect to see the results of RT in the revision.

Authors: Response time, though we agree is an important topic, was not reported because it was not the subject of the investigation, which was strictly focused on recall accuracy. The data around recall time was not retrievable and unfortunately could not be added retrospectively.

There are precedents in published work around n-back test and tDCS not focusing on response time such as the papers below. 

• Mylius, V., Jung, M., Menzler, K., Haag, A., Khader, P.H., Oertel, W.H., Rosenow, F. and Lefaucheur, J.P., 2012. Effects of transcranial direct current stimulation on pain perception and working memory. European journal of pain, 16(7), pp.974-982.

• Cheng, C.P., Chan, S.S., Mak, A.D., Chan, W.C., Cheng, S.T., Shi, L., Wang, D. and Lam, L.C.W., 2015. Would transcranial direct current stimulation (tDCS) enhance the effects of working memory training in older adults with mild neurocognitive disorder due to Alzheimer’s disease: study protocol for a randomized controlled trial. Trials, 16(1), p.479.

3.6 I find one subject’s data is abnormal (below the chance level of 0.5) in the Fig 2A. Please provide the other version of the statistical results in the revision where you should delete this data point to find out if it affects the current results. 

Authors: The Y-axis in Figure 2A is refers to accuracy scores (not probabilities. We are happy to provide the results without this subject. 

The results are recomputed after removing the outlier subject using Two-way ANOVA results are as follows:

• Stimulation (sham and real): F(1,76)=3.91, p=0.05, partial eta=0.05

• Stimuli (letters and alphabets): F(1,76)=5.17, p=0.02, partial eta=0.06

• Interaction effect: F(1,76)=0.068, p=0.8, partial eta=0.001

• Two-tail paired t-test between sham and tDCS across shapes: 0.02 (Cohen’s d=0.56)

• Two-tail paired t-test between sham and tDCS across letters: 0.15 (Cohen’s d=0.36)

We have presented the non-altered data and altered data (after removing outlier) in the paper for completeness. 

3.7 The interaction effect was not significant, so why did you conduct a simple effect analysis by t-test? This doing seems not follow statistical requirements. 

Authors: Our understanding is, if there is a significant interaction effect, then the post-hoc on the main effects are often not of interest. However, the post-hoc on the interaction is of interest.

But if there isn’t a significant interaction effect, but there are significant main effects, then the post-hoc is performed on the significant main effects.

A number of papers indicate this approach: Wei et al (“Comparisons of treatment means when factors do not interact in two-factorial studies”, Amino Acids, 2012 42(5):2031-5) provide examples of 2-factor studies where it is useful to perform a post-hoc analysis when only one factor and not both factors or their interaction is significant. In the abstract, they wrote "when the two factors do not interact, a common understanding among biologists is that comparisons among treatment means cannot or should not be made. Here, we bring this misconception into the attention of researchers. "

3.8 The df and t values are wrong (t(38)=-2.81, p=0.011...).

Authors: We have now addressed this comment in page 7, line 21. 

4.1 . Discussion. Not using 0-back as the control condition should be the other one limitation. As far as know, 0-back is the control condition for the classical n-back task. Why not use it in your study? Else, please discuss the influence of task difficulty between these two tests on your results.

Authors: Our understanding is 0-back test means no recall involved/comparison between current cue and past ones in n steps back stimuli (if n=0). We can see how this can be used as a baseline to ensure participants are at least able to operate the experiment (click when they see the cues). 

N-back test as a working memory is obviously useful from n=1 and higher. We did experiment in pilot experiment, with values of n and yes, test difficulty increases with increase in n value. We looked at the literature the highest “n” used to the authors knowledge (Ohn eta l., 2007; Frengi et al., 2005; Nilson et al., 2015) is 3 and lowest is 1 (Jonides et al., 1997; Ragland eta l., 2002; Carlson et al., 1998). 

Prior to starting this experiment, the question of what is the appropriate value for n was asked, and the team did run a smaller pilot with n values (1, 2 and 3). For n=1 accuracy was close to 100% (too easy) , whereas n=3 was difficult for subjects with accuracy mostly close to 0% (too hard). 2-back was found to provide the most range of accuracy results and hence was chosen for the main experiment. 

We have now added the above statement in the methods section (page 6, line 4-7).

5.1 Reference. The order of 54 and 55 is wrong.

Authors: Unless we are misunderstanding the reviewer’s comment, the order seems to be correct. Ref 54 talks about repeated A-tdcs sessions and Ref 55 talks about steroid hormones effecting the mood and concentration level. 

 6. Figure 1. The rectangle in the top left corner of Fig 1 seems to be unnecessary. The red line under the “tDCS” should be removed.

Author: This comment has now been addressed. 

We thank again the reviewer for his/her time examining our manuscript, and we remain happy to give any further details and clarification if needed before the final publication of the paper.

---

## [Decision Letter · Decision Letter 1]

10 Mar 2020

PONE-D-19-24574R1

­­Transcranial Direct Current Stimulation and Working Memory: Comparison of effect on Learning Shapes and English Letters

PLOS ONE

Dear Dr Ramaraju,

Thank you for submitting your manuscript to PLOS ONE. After careful consideration, we feel that it has merit but does not fully meet PLOS ONE’s publication criteria as it currently stands. Therefore, we invite you to submit a revised version of the manuscript that addresses the points raised during the review process.

We would appreciate receiving your revised manuscript by Apr 24 2020 11:59PM. To enhance the reproducibility of your results, we recommend that if applicable you deposit your laboratory protocols in protocols.io, where a protocol can be assigned its own identifier (DOI) such that it can be cited independently in the future. For instructions see: http://journals.plos.org/plosone/s/submission-guidelines#loc-laboratory-protocols

We look forward to receiving your revised manuscript.

Kind regards,

Tifei Yuan

Academic Editor

PLOS ONE

Reviewers' comments:

Reviewer's Responses to Questions

**Comments to the Author**

1. If the authors have adequately addressed your comments raised in a previous round of review and you feel that this manuscript is now acceptable for publication, you may indicate that here to bypass the “Comments to the Author” section, enter your conflict of interest statement in the “Confidential to Editor” section, and submit your "Accept" recommendation.

Reviewer #1: (No Response)

2. Is the manuscript technically sound, and do the data support the conclusions?

Reviewer #1: Partly

3. Has the statistical analysis been performed appropriately and rigorously? 

Reviewer #1: No

4. Have the authors made all data underlying the findings in their manuscript fully available?

Reviewer #1: Yes

5. Is the manuscript presented in an intelligible fashion and written in standard English?

Reviewer #1: Yes

6. Review Comments to the Author

Reviewer #1: The author has responded to and revised some of the comments. However, I still think that some problems have not been directly explained by the authors. First, there is a lack of baseline condition (0-back). Second, no RT results were presented. Both of these are very important for experimental design and consequence inference.

7. PLOS authors have the option to publish the peer review history of their article (what does this mean?). If published, this will include your full peer review and any attached files.

Reviewer #1: No

---

## [Author Response · Author response to Decision Letter 1]

11 May 2020

Response to Reviewers comments

Reviewer #1: The author has responded to and revised some of the comments. However, I still think that some problems have not been directly explained by the authors. First, there is a lack of baseline condition (0-back). Second, no RT results were presented. Both of these are very important for experimental design and consequence inference.

We thank the reviewer again for their constructive comments and for acknowledging that most issues have been addressed with the exception of only two remaining questions, around use of 0-back test and, the reaction time measurement which we accept are fair, and legitimate points and shall attempt to address below.

0-back test: 

Upon further consideration of this concept, we understand the reviewer’s point as is referring to a baseline condition indicating that the subjects recognised the presence of an appropriate symbol and then reacted with no demand on memory, as in the reference below:

"In the 0-back condition, the target was any letter that matched a pre-specified letter (i.e., “c”). Thus, this condition required sustained attention but no working memory demand." p.712 (Miller et al, 2009)

In this case, this was performed, though admittedly not a formal part of the study, as part of the pre-study pilot we referred to in the paper were subjects were asked to accustom themselves to the test and as part of that process, respond to visual letter stimuli presented. The accuracy rate for that was 100%. We did not label it as a 0-back test as it was seen as simply testing that baseline condition and ensuring subject were “capable” of performing the main 2-back test. We did not formally test the sustained attention element, however as each test period on a single stimulus type was circa 3 minutes, attrition was not considered an important factor. 

We have now added this important clarification to the paper in (page 6-line 4).

As we mentioned in our previous response we went further in the pilot, performing the 1-back and 3back tests in the pilot; however decided not to pursue these tests for the main experiment as the dynamic variation for each was small (all results being near 100% or near 0% accuracy, for the 1-back and 3 back respectively).

Reaction time: 

Here also, we understand the important of response time in research aimed at studying short term memory. We agree with the reviewer that RT is essential before drawing any conclusions about mechanisms of any reported effects on the memory. We have added this line to our paper Page 8, Line 39 when discussion the limitations of the study:

“Finally, the study did not look at “reaction time” and its interaction with accuracy and whether a ceiling effect (Hur et. al. 2017) could have been affected by tDCS application.”

This being said, none inclusion of response time appears to be common when the target of study is not to understanding memory per say but the effect of a factor (in our case tDCS) on accuracy specifically. Examples of recent such peer reviewed research work below are 

• Scharinger, C., Soutschek, A., Schubert, T. and Gerjets, P., 2017. Comparison of the working memory load in n-back and working memory span tasks by means of EEG frequency band power and P300 amplitude. Frontiers in human neuroscience, 11, p.6.

• Soveri, A., Antfolk, J., Karlsson, L., Salo, B. and Laine, M., 2017. Working memory training revisited: A multi-level meta-analysis of n-back training studies. Psychonomic bulletin & review, 24(4), pp.1077-1096.

• Yaple, Z. and Arsalidou, M., 2018. N‐back working memory task: Meta‐analysis of normative fMRI studies with children. Child development, 89(6), pp.2010-2022.

We hope our acknowledgment of the RT limitation in the paper, along with the fact paper is not at complete odds with similar peer reviewed research, will meet the reviewer’s, if not full satisfaction, minimal approval for accepting this work.

Best regards,

Authoring teams

---

## [Decision Letter · Decision Letter 2]

18 May 2020

PONE-D-19-24574R2

­­Transcranial Direct Current Stimulation and Working Memory: Comparison of effect on Learning Shapes and English Letters

PLOS ONE

Dear Dr Ramaraju,

Thank you for submitting your manuscript to PLOS ONE. After careful consideration, we feel that it has merit but does not fully meet PLOS ONE’s publication criteria as it currently stands. Therefore, we invite you to submit a revised version of the manuscript that addresses the points raised during the review process.

We would appreciate receiving your revised manuscript by Jul 02 2020 11:59PM. To enhance the reproducibility of your results, we recommend that if applicable you deposit your laboratory protocols in protocols.io, where a protocol can be assigned its own identifier (DOI) such that it can be cited independently in the future. For instructions see: http://journals.plos.org/plosone/s/submission-guidelines#loc-laboratory-protocols

We look forward to receiving your revised manuscript.

Kind regards,

Tifei Yuan

Academic Editor

PLOS ONE

Reviewers' comments:

Reviewer's Responses to Questions

**Comments to the Author**

1. If the authors have adequately addressed your comments raised in a previous round of review and you feel that this manuscript is now acceptable for publication, you may indicate that here to bypass the “Comments to the Author” section, enter your conflict of interest statement in the “Confidential to Editor” section, and submit your "Accept" recommendation.

Reviewer #1: All comments have been addressed

2. Is the manuscript technically sound, and do the data support the conclusions?

Reviewer #1: Yes

3. Has the statistical analysis been performed appropriately and rigorously? 

Reviewer #1: Yes

4. Have the authors made all data underlying the findings in their manuscript fully available?

Reviewer #1: No

5. Is the manuscript presented in an intelligible fashion and written in standard English?

Reviewer #1: Yes

6. Review Comments to the Author

Reviewer #1: Minor edit to address before endorsing publication:

The df of t tests in Page 7, Lines 20 and 21 should be 19. Please check and modify it.

7. PLOS authors have the option to publish the peer review history of their article (what does this mean?). If published, this will include your full peer review and any attached files.

Reviewer #1: No

---

## [Author Response · Author response to Decision Letter 2]

20 May 2020

Reviewer #1: Minor edit to address before endorsing publication:

The df of t tests in Page 7, Lines 20 and 21 should be 19. Please check and modify it.

We have now addressed the final comment of the reviewer. The updated values can be found page 7, lines 15 and 16. 

We thank the reviewer again for their constructive comments which have invariably improved the quality of our paper.

---

## [Decision Letter · Decision Letter 3]

26 Jun 2020

­­Transcranial Direct Current Stimulation and Working Memory: Comparison of effect on Learning Shapes and English Letters

PONE-D-19-24574R3

Dear Dr. Ramaraju,

We’re pleased to inform you that your manuscript has been judged scientifically suitable for publication and will be formally accepted for publication once it meets all outstanding technical requirements.

Kind regards,

Tifei Yuan

Academic Editor

PLOS ONE

Additional Editor Comments (optional):

Reviewers' comments:

Reviewer's Responses to Questions

**Comments to the Author**

1. If the authors have adequately addressed your comments raised in a previous round of review and you feel that this manuscript is now acceptable for publication, you may indicate that here to bypass the “Comments to the Author” section, enter your conflict of interest statement in the “Confidential to Editor” section, and submit your "Accept" recommendation.

Reviewer #1: All comments have been addressed

2. Is the manuscript technically sound, and do the data support the conclusions?

Reviewer #1: Yes

3. Has the statistical analysis been performed appropriately and rigorously? 

Reviewer #1: Yes

4. Have the authors made all data underlying the findings in their manuscript fully available?

Reviewer #1: No

5. Is the manuscript presented in an intelligible fashion and written in standard English?

Reviewer #1: Yes

6. Review Comments to the Author

Reviewer #1: The authors have completed all revisions.

I endorse this publication.

7. PLOS authors have the option to publish the peer review history of their article (what does this mean?). If published, this will include your full peer review and any attached files.

Reviewer #1: No

---

## [Editor Report · Acceptance letter]

14 Jul 2020

PONE-D-19-24574R3 

­­Transcranial Direct Current Stimulation and Working Memory: Comparison of effect on Learning Shapes and English Letters 

Dear Dr. Ramaraju:

I'm pleased to inform you that your manuscript has been deemed suitable for publication in PLOS ONE. Congratulations! Your manuscript is now with our production department. 

Kind regards, 

on behalf of

Dr. Tifei Yuan 

Academic Editor

PLOS ONE